# Horizontal Gene Transfer of Antibiotic Resistance Genes in Biofilms

**DOI:** 10.3390/antibiotics12020328

**Published:** 2023-02-04

**Authors:** Claudia Michaelis, Elisabeth Grohmann

**Affiliations:** Faculty of Life Sciences and Technology, Department of Microbiology, Berliner Hochschule für Technik, 13347 Berlin, Germany

**Keywords:** antibiotic resistance, antibiotic resistance genes, horizontal gene transfer, conjugation, biofilm, transduction, transformation, plasmids, fluorescence microscopy

## Abstract

Most bacteria attach to biotic or abiotic surfaces and are embedded in a complex matrix which is known as biofilm. Biofilm formation is especially worrisome in clinical settings as it hinders the treatment of infections with antibiotics due to the facilitated acquisition of antibiotic resistance genes (ARGs). Environmental settings are now considered as pivotal for driving biofilm formation, biofilm-mediated antibiotic resistance development and dissemination. Several studies have demonstrated that environmental biofilms can be hotspots for the dissemination of ARGs. These genes can be encoded on mobile genetic elements (MGEs) such as conjugative and mobilizable plasmids or integrative and conjugative elements (ICEs). ARGs can be rapidly transferred through horizontal gene transfer (HGT) which has been shown to occur more frequently in biofilms than in planktonic cultures. Biofilm models are promising tools to mimic natural biofilms to study the dissemination of ARGs via HGT. This review summarizes the state-of-the-art of biofilm studies and the techniques that visualize the three main HGT mechanisms in biofilms: transformation, transduction, and conjugation.

## 1. Introduction

Up to 80% of all bacteria mainly live in biofilm communities [1,2]. The natural habitats of biofilms include aquatic environments (rivers, hot springs, or streams), soil and rhizosphere, sediment, and gastrointestinal surfaces, or they occur as dental plaque [2,3,4,5,6,7]. The definition of biofilms is not restricted to microbial layers on surfaces. Biofilms form at any liquid–liquid, solid–liquid, solid–gas, liquid–gas, or air–air interfaces (”bubble biofilm”) [8,9,10]. Furthermore, the biofilm state applies to multicellular (non-attached) microbial aggregates such as flocs, microlayers, endolithic microbial populations, pellicles, slimes, microbial mats, microorganisms in soils, sediments and aerosols, marine snow, sludge, or biofilm on leaves and plant roots [10,11].

Although biofilms can be beneficial for gut homeostasis and food digestion [12], environmental bioremediation [13] and biotechnological applications [14], they are also responsible for microbial corrosion [15] and biofouling [16]. In addition, they are associated with serious diseases and infections. The global economic impact of biofilms is estimated to cost USD 5000 billion annually [17]. For this reason, studying bacterial biofilm formation has gained much attention in recent decades [18]. Several in vivo and in vitro biofilm models have been developed to mimic natural biofilms. Testing under biofilm conditions, rather than in the planktonic state, is necessary to develop therapeutic agents for biofilm-forming bacterial pathogens. Thus, most biofilm studies have been performed to test drug susceptibility or to develop treatment strategies for chronically infected wounds, implant and device-related infections, or biofilm-associated infections [18,19]. Biofilms are also of primordial interest as hotspots for horizontal gene transfer (HGT) and therefore for the dissemination of antibiotic resistance genes (ARGs). Studies on environmental biofilms are not as numerous by far as those on health-related bacteria.

### 1.1. Methodology

This review presents the state-of-the-art of the literature on HGT in biofilms. We aimed to show HGT studies of both Gram-positive and Gram-negative bacteria in biofilms. However, research on conjugative plasmids and ICEs from Gram-positive bacteria has been highlighted, as it has not been reviewed as extensively as works from Gram-negative bacteria. Furthermore, we wanted to point out the drawbacks and restrictions of HGT studies in biofilm communities.

PubMed and Google scholar were used as electronic databases for the literature search.

The keywords used in the bibliographic research were: “abiotic factor”, “antibiofilm”, “antibiotic resistance”, “biofilm”, “biofilm formation”, “biofilm formers”, “biofilm study”, “biofilm-associated infection”, “biotic factor”, “CLSM”, “conjugation”, “conjugative plasmid”, “device-related infection”, “environment”, environmental factor”, “horizontal gene transfer”, “hospital”, “hot spot”, “ICE”, “IME”, “implant”, “*in situ* model”, “in vitro model”, “in vivo model”, “integrative and mobilizable elements”, “integrative conjugative elements”, “mating”, “medical device”, “membrane vesicle”, “microfluidic”, “monomicrobial”, “mono-species”, “multi-drug resistance”, “multi-species”, “nosocomial infection”, “outer membrane vesicle”, “pathogen”, “phage“, “phage therapy”, “planktonic culture”, “plasmid”, “polymicrobial”; “T4SS”, “transduction”, “transformation”, “transposon” or “wastewater treatment plant”.

The selection of articles is based on the following main criteria: (1) experimental evidence for HGT and (2) “real” biofilm studies.

### 1.2. Biofilm Development and Occurrence

The multistep biofilm formation process has been studied in great detail. It starts with the adherence of planktonic cells to a surface and EPS formation, followed by microcolony formation, differentiation into a mature biofilm and partial detachment and dispersion for new surface colonization [11,20,21]. The four main stages of biofilm formation include the initial attachment of planktonic cells to a solid surface, subsequent adhesion, biofilm maturation and dispersion of planktonic cells from a mature biofilm (Figure 1).

Since the biofilm state applies to several biofilm architectures [10], the developmental stages of biofilm formation cannot be generalized for non-surface-attached biofilms. The original concept of a biofilm life cycle is based on *in vitro Pseudomonas aeruginosa* biofilms and has been widely generalized to describe all biofilms [22]. However, the conceptual model cannot be equally applied to in vivo, in vitro and *in situ* conditions of biofilm growth. A recent review critically pointed out the limitations of the classical multistep biofilm model and proposed an updated, broader model of biofilm formation [11]. The model includes three main steps such as (1) aggregation and attachment, (2) growth and accumulation, and (3) disaggregation and detachment.

Biofilms can consist of single or multiple species, the latter being the predominant form in nature [23,24,25]. Biofilm-forming bacteria are embedded in self-secreted extracellular polymeric substances (EPS) to protect themselves in hostile environments, in the presence of antimicrobials or to circumvent host defenses [26,27,28]. They are primarily composed of water, enzymes, extracellular DNA (eDNA), polysaccharides, and lipids [29,30].

Artificial surfaces such as medical devices, wastewater pipelines, food-processing surfaces, equipment in the dairy, wine, and textile industries, and microplastics are commonly colonized by biofilms [27,31,32,33,34,35].

These are often formed by multiple antibiotic-resistant Gram-negative human pathogens such as *Acinetobacter baumanii*, *Klebsiella* spp., *Escherichia coli*, *P. aeruginosa*, *Proteus mirabilis*, *Serratia marcescens*, *Enterobacter cloaceae* and *Burkholderia cepacia* [36,37,38,39,40]. Food-borne biofilm-forming pathogens include *E. coli*, *Salmonella* spp., *Serratia* spp., *Listeria monocytogenes*, *Vibrio* spp. and *Campylobacter jejuni* [41,42,43,44,45].

Important Gram-positive biofilm producers include healthcare-associated pathogens such as *Streptococcus* spp., *Enterococcus* spp., and *Staphylococcus* spp. Coagulase-negative staphylococci (CoNS) and *Staphylococcus aureus* are the leading causes of biofilm-associated infections [39,46,47,48,49].

*Candida albicans* also forms biofilms. The fungal pathogen often co-colonizes with bacterial pathogens and contributes to biofilm-associated infections [50,51,52,53].

Nosocomial infections such as pneumonia, urinary tract infections or device-related infections are predominantly caused by biofilm producers [54]. According to the National Institutes of Health (NIH) and Centers for Disease Control and Prevention (CDC), more than 60% of nosocomial infections and up to 80% of microbial infections are due to biofilms [55]. In the last few years, the spread of biofilms on medical implants and devices in clinical settings was the subject of many studies and has been frequently reviewed [56,57,58,59]. Medical devices can be reservoirs for multidrug-resistant, biofilm-producing strains [39,60,61].

### 1.3. Biofilms as Reservoir for Antibiotic Resistance Genes

Wastewater and soil are major reservoirs for antibiotic resistomes [62]. The addition of antibiotic-containing manure, sewage sludge or other pollutants such as heavy metals, residuals of veterinary antibiotic compounds to agricultural soils can lead to increasing levels of antibiotic resistance and antibiotic-resistant bacteria [63,64,65]. Organic fertilizers have been shown to contain various human pathogenic bacteria [66]. Some of them survive in the environment, colonize the soil and become enriched in the rhizosphere [67]. Subinhibitory concentrations of antibiotics can promote HGT in soil, and therefore, the selection of resistant pathogens. Soil-indwelling pathogens may recolonize the human gut when ingested by humans, leading to foodborne illnesses due to the consumption of contaminated food [68,69].

On the other hand, the spread of multiple ARGs among clinically relevant pathogens is a global threat that fosters mortality and poses an enormous economic burden [70]. The inadequate use and inappropriate prescription of antibiotics have led to a global antibiotic resistance crisis. The increase in multidrug-resistant bacterial infections and unsuccessful antibiotic treatments underlines the fact that ARGs cannot be ignored. In particular, biofilm formation promotes important features of pathogenesis such as significantly reduced susceptibility to antibiotics compared to susceptible non-biofilm producers [26,71,72,73].

Antimicrobial susceptibility tests on clinical isolates are normally performed using planktonic cultures. However, biofilms respond differently to antibiotic dosage which could lead to inadequate treatment and even treatment failure. A recent study from Lozano et al. described a simple biofilm model of human sputum to test the sensibility of *P. aeruginosa* strains against different antibiotics [72]. The development of specific growth media to better mimic in vivo biofilm formation and new biofilm antimicrobial susceptibility tests could contribute to the reduction in treatment failure [74,75,76].

There is an urgent need to develop new antibiotics and strategies to prevent the development of antibiotic resistance [77]. It is not only clinical environments that support the spread of ARGs, but wastewater treatment plants (WWTPs) and agriculture are also considered hotspots for ARG propagation [78,79,80,81]. If ARGs are not completely removed by wastewater treatment, they are released into the wastewater effluent. This contributes to a vicious cycle of ARG dissemination in the environment [82].

Biofilms are considered as a driving force for ARG dissemination because the proximity of bacteria in a biofilm enhances cell-to-cell contact and, therefore, the probability of genetic exchange [83,84].

### 1.4. Horizontal Gene Transfer in Biofilms

Biofilms facilitate the exchange and recycling of nucleic acids due to the proximity and long retention time of cells within the EPS matrix [10]. Constantly changing environments shape the genetic composition of bacteria via various mechanisms. Point mutations or recombination alter the bacterial genome and are the first steps in genetic diversification. Genetic material is transmitted via vertical transmission as well as with HGT. This mechanism can transform harmless bacteria into major human pathogens. HGT includes DNA transfer from a donor cell to a recipient cell via direct cell-to-cell contact (conjugation), uptake of extracellular naked DNA by competent cells (transformation), or bacteriophage-mediated DNA transfer (transduction).

A fourth mechanism describes the outer membrane vesicle (OMV)-mediated transfer of genetic material via cell fusion. It is less well described in the literature compared to other HGT mechanisms [85,86,87]. Membrane vesicles originating from Gram-positive bacteria are called membrane vesicles (MVs), and those originating from Gram-negative bacteria are called OMVs [88]. OMVs can act as gene transfer vectors. They can contain different types of genetic material such as plasmid DNA, chromosomal DNA, phage DNA, or RNA [85]. The way in which OMVs contribute to HGT has recently been reviewed [87]. OMV-mediated transfer has been shown for several biofilm-forming organisms, such as *E. coli*, *Acinetobacter baylyi, P. aeruginosa*, *A. baumanii* or *Porphyromonoas gingivalis*. A recent study has shown that OMVs secreted by *K. pneumoniae* can act as cargo for ARGs [89]. Two different β-lactam resistance plasmids were incorporated into OMVs and transferred among phylogenetically distant bacteria such as *E. coli*, *S. enterica*, *P. aeruginosa*, and *B. cepacia*. Furthermore, *A. baumannii* OMVs can secrete the plasmid-borne *bla*_OXA-24_ gene [90] and *E. coli* OMV plasmids containing *bla*_CTX-M-55_ [91]. Little is known about how MV-mediated HGT occurs in Gram-positive bacteria [88,89,92].

The extent to which OMV-mediated transfer contributes to the dissemination of ARGs in biofilms is not known as most studies on OMVs have been conducted on planktonic cultures.

OMVs have been shown to promote biofilm formation in *Aeromonas* spp. [93], *P. aeruginosa* [94,95]*, Pseudomonas putida* [96], *Helicobacter pylori* [97], and *Vibrio cholerae* [98]. On the other hand, several studies have demonstrated that OMVs and MVs can act as antibiofilm agents, such as *Burkholderia thailandensis* MVs for *Streptococcus mutans* biofilms [99]. *P. aeruginosa* OMVs were shown to be involved in growth inhibition of *S. aureus* in a dual-species biofilm [94].

DNA encapsulated into OMVs is protected from exonuclease degradation, inactivation via freezing, and thermodegradation [100]. These environmentally inert vehicles are of particular interest for OMV-mediated HGT in biofilms and harsh environments such as hot springs or the Antarctic.

Another mode of HGT is mediated by nanotubes that were discovered in *B. subtilis* [101,102]. These membranous, elongated extracellular structures enable the connection of neighboring cells in proximity. This mediates the contact-dependent exchange of molecules between the cells. Their impact and physiological relevance in shaping genetic diversity in biofilms has not yet been explored.

HGT research originated in the late 1920s. When DNA was determined to be the source for transfer, the repertoire of methods to examine gene transfer expanded. HGT occurs within the same species, but also crosses phylogenetic barriers thereby setting no limits to acquiring resistance from phylogenetically unrelated bacteria from diverse environmental habitats [103,104]. The driving factors of HGT are mobile genetic elements (MGEs) such as plasmids, ICEs, integrons, or transposons [84,105,106,107]. As most bacteria live in biofilms in nature, it seems reasonable that HGT occurs more frequently in biofilms than between planktonic cells [83,108].

### 1.5. Environmental Factors Influencing Biofilm Formation

Although implant materials are selected based on their optimal biocompatibility and treatment success, they unintentionally promote biofilm formation and can lead to biofilm-associated infections. The accumulation of biofilm producers and transmission of foodborne pathogens to humans via food-processing surfaces during processing, storage, production, or packaging are serious problems [109].

Common materials for implants are metals such as stainless steel or titanium, ceramics, and polymers such as polyurethane or silicone. Surface parameters and material properties influence microbial adhesion [110,111,112]. Modification of the surface topography [110,113] and development of antimicrobial implant coatings [114,115] are efficient approaches for inhibiting biofilm formation and preventing biofilm-associated infections.

Several recent reviews have described further variables that influence biofilm formation in detail. Temperature, pH, water, nutrient and oxygen availability, ion content, protozoa, salinity, or exopolysaccharide release by OMVs all affect biofilm formation [10,116,117,118].

Biofilm formation is affected not only by environmental factors, but also by the EPS composition of biofilms. A study from Blanco et al. demonstrated that EPS composition is strongly influenced by environmental parameters and plays a critical role in ecological adaptation to extreme environments [119].

Biofilms undergo shear stress and different fluid dynamics in the human body (veins, catheter environments) as well as in aquatic systems [111,112,120]. Shear dynamics further influence the composition of multispecies biofilms [112,121].

Sub-minimum inhibitory antibiotic concentrations (sub-MICs) are concentrations lower than MIC values. They can lead to antibiotic-induced stress and promote biofilm formation [60,122].

### 1.6. Abiotic and Biotic Factors Influence Horizontal Gene Transfer in Biofilms

Plasmid transfer rates in biofilms are affected by environmental abiotic factors such as oxygen concentration and availability [123,124] or stimulatory effects of subinhibitory antibiotic concentrations [125,126]. Both the presence of mature biofilms [127,128] and cell density affect horizontal transfer of MGEs in biofilms.

In addition to abiotic factors, many biotic factors have been discussed to influence the spread: co-residing (conjugative) plasmids or virulence determinants in the donor or recipient cell [129,130], plasmid incompatibility [129], or genetic compatibility of the MGE with the host.

Moreover, the copy number of the (conjugative) plasmid can be increased during biofilm growth compared to their planktonic counterparts, independent of their replicons and replication mechanisms [131,132]. Other parameters include host factors such as secretion of immunoglobulins or proteases for biofilms on eukaryotic tissues [12], the taxonomic relatedness between donor and recipient [133], and plasmid–donor–recipient-combinations [134]. In absence of selection, it has been shown that conjugative plasmids were retained for longer in biofilms compared to planktonic growth, and thus acted as plasmid sinks [135,136].

Conjugation can be affected by the biofilm composition and the design of the mating experiment: (I) only one strain (donor or recipient) is allowed to attach to the surface, and the second strain is added to the pre-developed biofilm layer [137,138] or (II) a defined mixture of donor and recipient strains is inoculated from the beginning [127,128,139,140,141,142]. In addition, there is no standard method for determining the conjugative transfer rate. Therefore, experimental results are difficult to compare. Furthermore, terminology varies greatly in terms of how the rate of HGT in biofilms is described. This review will use the following terms to quantify the gene spread via the HGT mechanisms, transformation, transduction, and conjugation: transformation rate, transduction rate, and conjugative transfer rate.

The determination of conjugative transfer rates can be biased by several experimental conditions such as the growth rate of donor and recipient strains, initial population densities, or donor-to-recipient ratio. To bypass biased experimental results through (a)biotic variations, a computational tool was developed to allow accurate and comparable measurements of conjugative plasmid transfer rates in liquid matings [143]. Developing similar mathematical models for determining conjugative transfer rates in biofilms remains a challenge. Ensuring comparability across experimental conditions should account for the spatial structure of biofilms.

### 1.7. State of the Art of Biofilm Studies

Our understanding of HGT in clinical and environmental settings is based primarily on in vitro studies. Single-species and planktonic cultures can be better cultivated under optimized in vitro laboratory conditions in broth or on agar surfaces. This kind of homogenous cultivation might not reflect natural features and has limitations, such as the lack of a three-dimensional architecture and high cell density of biofilms [144]. Another key difference between in vitro and in vivo models is the absence of a host immune system in in vitro biofilms [145]. Although the cultivation of bacterial colonies on agar plates should be much closer to biofilm growth, several conditions such as the absence of shear stress or the method of nutrient supply can fundamentally differ from those of natural biofilms.

Other mechanisms that might influence HGT in biofilms include (I) limited interactions between donor and recipient cells in a spatially structured population creating subpopulations, (II) regulatory control of the conjugative process, (III) environmental stressors (toxins, antibiotics, and UV irradiation), (IV) nutrient supply, and (V) cell density [134].

As mixed-species biofilms and polymicrobial infections are much more common in nature [37,146,147], biofilm studies with multiple species should be performed. Clinical isolates from medical devices have shown that biofilm infections are not only due to single species but also to polymicrobial colonization including even copresence of fungi and bacteria [39].

A better understanding of HGT mechanisms in biofilms would provide more clarity regarding the rapid adaptation of pathogens to changing environmental conditions and the development of multi-resistant species.

However, researchers must be aware that in vitro models do not accurately mimic the human host or the natural environment. To date, most biofilm studies have been performed in vitro, owing to the possibility of high-throughput testing, easy setup, and low cost [19,145]. Several in vitro and in vivo models have been extensively reviewed [19,145,148,149]. Despite the availability of numerous methods for developing biofilm models, such as microtiter plate assays and static or fluid cultivation, mimicking complex biofilms remains technically challenging. The cultivation model of choice strongly depends on the research objective and what is intended to be simulated. When selecting a biofilm model, caution is needed when interpreting test results because of in vivo/in vitro differences in biofilm biology [150]. Furthermore, there is a risk of oversimplifying models when using standard growth media or abiotic surfaces [19].

Biofilm characterization has mainly focused on clinical isolates or anthropogenic systems such as WWTPs. Studies on biofilms from natural environments have been mostly performed with environmental isolates rather than with *in situ* biofilms. Research on *in situ* and in vivo biofilms remains time-consuming and expensive. Some studies have used non-mammalian models as alternative hosts, such as *Drosophila melanogaster* [151,152] and *Caenorhabditis elegans* [153] to investigate biofilm infections and intestinal colonization *in vivo.*

### 1.8. Culture-Independent Approaches

Most biofilm studies have been conducted using characterized laboratory strains. A stronger focus should be laid on plasmids and strains of both environmental and clinical origin. These studies could help to mimic clinical and environmental scenarios.

Not only cultivation-based approaches are required to study the mobilization and transfer of ARGs; culture-independent techniques, such as flow cytometry, allow the detection and quantification of transconjugants (recipients which have acquired a plasmid via conjugation) within biofilms [141,154,155] without plate counting. Phenotypically trackable markers such as fluorescently labeled plasmids can be detected with flow cytometry or fluorescence microscopy.

Quantitative PCR (qPCR) is another classical culture-independent methodology to detect ARGs, their abundance, and distribution. However, it is not suited to determine transfer rates. Metagenomic data contribute to the prediction of HGT events in microbial communities [83]. These analyses can also bypass cultivation and provide an overview of the distribution and occurrence of ARGs, bacteria, and MGEs in a habitat of interest. The major bottleneck is the lack of reliable assignment of ARGs to their origin in natural communities (MGEs or bacterial chromosomes) when using standard metagenomic assemblies such as shotgun sequencing [156,157].

CRISPR–Cas spacer acquisition is another technique enabling real-time recording of HGT events at nucleotide resolution and thus, level of DNA acquisition [158]. Spacer acquisition describes the incorporation of small fragments of foreign DNA, known as spacers, into CRISPR loci [159]. An engineered *E. coli* recipient (“recording”) strain contains a plasmid with the *cas1*-*cas2* operon whereby mobile DNA entering the cell is captured and incorporated into the CRISPR array. Transient HGT events such as those lacking replication of MGEs or those with low transfer frequencies can be studied using this approach. However, the recording system is still restricted to *E. coli.*

The in vivo proximity-ligation method Hi-C [160] enables the correct linkage between MGEs, such as plasmids and phages, to their hosts. Thus, the *in situ* host ranges of MGEs and the origin of ARGs can be tracked in environmental microbial communities without cultivation [161,162]. It would be very beneficial to understand to which hosts or MGEs ARGs belong to, and which plasmids interact with which hosts. Hi-C metagenomics broadens our knowledge of plasmid–host interactions in large, natural communities [82,157].

On this basis, combinations of microbial cultivation, in vivo studies with *Drosophila* or *C. elegans*, and *in situ* biofilm studies can contribute to a more complete understanding of HGT in biofilms.

The following chapters focus on the three main HGT mechanisms—transformation, transduction, and conjugation—and how they have been examined in biofilms. Figure 1 illustrates these three main HGT mechanisms occurring in polymicrobial biofilms.

## 2. Transformation

Natural transformation (also known as natural competence) was discovered in 1928 as the first mechanism of HGT which does not rely on MGEs. Transformation largely depends on the uptake and incorporation of exogenous naked DNA from the environment into the genomes of competent recipient organisms [163,164]. Internalized genetic material is permanently incorporated into the bacterial chromosome by homologous recombination or serves as a nutrient source, bypassing de novo biosynthesis of nucleotides [165]. Competence is a developmental state that occurs during specific growth phases, except for *H. pylori* and *Neisseria* spp., which are likely to be constitutively competent [166]. Cells can stop growing (growth arrest) when in a competent state and express proteins for DNA import [167,168].

In contrast to conjugation and transduction, transformation relies only on the recipient bacterium that expresses the competence machinery (Figure 1). Competence regulation is based on several conserved competence-inducing genes and quorum-sensing (QS) systems involved in exogenous DNA uptake and integration into the chromosome [165,169,170]. Fluorescence-based and qPCR-based studies have shown that competence genes are upregulated in *S. aureus* [171] and *Streptococcus pneumoniae* [172,173] under biofilm growth conditions compared with broth-grown bacteria.

Most transformable bacteria, both Gram-negative and Gram-positive, require type II-related secretion systems and type IV-like pilus structures for single-strand DNA (ssDNA) transport across the bacterial membrane [166,174].

Few transformation experiments have been conducted in biofilms compared with planktonic cultures. In the following sections, the methods by which transformation has been measured in biofilms are highlighted.

In 2017, the World Health Organization (WHO) published a list of priority pathogens for which the development of new antibiotics or other defense strategies is urgently required. The majority of these species are naturally competent [175,176]. Therefore, transformation can be a powerful tool for the distribution of ARGs. Natural transformation occurs in many bacteria (to date 82 species are known), including severe human pathogens such as *S. pneumoniae*, *V. cholerae*, and *H. pylori* [166]. Interestingly, naturally non-transformable pathogens, or pathogens for which natural transformation has not been demonstrated so far, can carry competence-related genes, such as streptococcal species other than *S. pneumoniae* [177], and *L. monocytogenes* [178,179]. However, transformation conditions and inducers of competence have not yet been elucidated. Thus, it is likely that the actual number of competent species is even higher.

Common model species used to study transformation are *V. cholerae*, *N. gonorrhoeae*, *Haemophilus influenzae*, *H. pylori*, *Acinetobacter* spp., and Gram-positive bacteria such as *B. subtilis* and *S. pneumoniae* [168].

### 2.1. Transformation of Gram-negative Bacteria in Biofilms

*V. cholerae* has long been considered as non-competent. The main niche for *V. cholerae* is the aquatic environment. It is often found in association with chitinous zooplankton, which induces competence in aquatic ecosystems. Therefore, this species can develop competence under special conditions when grown on chitin, as has also been shown for biofilms [180,181,182]. In all studies, *V. cholerae* cells were grown on sterile crab shell fragments submerged in artificial seawater. However, the cells were detached after incubation to determine the transformants on selective agar plates.

#### Methodology to Assess the Transformation Rate in Biofilms

A continuous flow chamber was used to co-culture two *N. gonorrhoeae* strains as a biofilm, and the transformation efficiency of chromosomally-encoded ARGs was compared to that of planktonic cells [183]. Transformation efficiency was higher in early biofilms than in planktonic cells but decreased in late biofilms. Cell aggregates were disrupted after biofilm removal from the chamber to determine the transformants on selective agar plates.

*P. aeruginosa* is known for its high genetic diversity and multidrug resistance. It has been recently shown that it can take up either genomic or plasmid DNA by natural transformation [184] in static broth assays and continuous flow biofilms [185]. Pioneering work by Hendrickx and coworkers proved natural competence in *A. baylyi* ADP1 (formerly named *Acinetobacter* sp. BD413) grown as monospecies biofilms in flow cells [186]. To study transformation, the *gfp*-carrying plasmid pGAR1 was used as the transforming DNA. In contrast to other studies, transformants were microscopically monitored in biofilms using GFP fluorescence and the nucleic acid stain Syto 60. Perumbakkan and coworkers compared the transformation of naturally competent single-species *A. baylyi* ADP1 biofilms with that of a soil biofilm community [187]. To this end, the broad-host-range cloning vector pBBR1MCS5 [188] was modified to carry the GFP reporter gene *gfpmut2* and an atrazine chlorohydrolase (*atzA*) gene for degradation of the herbicide atrazine. They examined the uptake and expression of the functional degradative gene in *Acinetobacter* biofilms in vivo as a potential technique for pollutant removal from bioreactors and aquifers. The biofilm was directly stained in the flow cell using Syto 60 and further examined using CLSM. Transformation frequency was determined by counting the total number of cells (fluorescence of Syto 60) and transformed cells (GFP fluorescence). The transformation frequencies obtained from both pure *Acinetobacter* biofilms and soil biofilms differed, depending on the biofilm sector corresponding to different biofilm heights. The authors assumed that biofilm height and shear forces influence transformation efficiency. The correlation between different biofilm metrics, such as density, cellular architecture, or EPS architecture (e.g., porosity or biovolume) and transformation frequency was demonstrated for monocultured *A. baylyi* BD4 and ADP1 biofilms under continuous flow conditions [189]. The biofilms were examined by CLSM for transformants using biofilm staining and GFP fluorescence (*gfp*-labeled non-conjugative pGAR38 plasmid as transforming DNA).

Santala et al. used a minimally invasive bioluminescence monitoring tool as a visualization technique for *in situ* transformations [190]. *A. baylyi* ADP1 was statically incubated in microtiter plates to obtain biofilm-like structures. Bioluminescence detection of transformation was performed using the bacterial luciferase operon of *Photorhabdus luminescens*. A replicative plasmid carrying the *luxCDABE* operon was used as DNA to be transformed. Homologous recombination was detected in a strain expressing *luxCDE* and introducing a non-replicative plasmid carrying an integrative *luxAB* gene cassette.

*E. coli* has to be artificially induced by electroporation or preparation of membranes for DNA uptake by treatment with calcium chloride or polyethylene glycol [191,192]. It was long thought that this species does not possess natural competence. However, several studies have shown natural competence in *E. coli* planktonic cells and *E. coli* biofilms under specific conditions [192,193,194,195,196].

### 2.2. Transformation of Gram-positive Bacteria in Biofilms

*B. subtilis* is one of the best-studied Gram-positive organisms and an important model system for studying biofilms [197]. However, only a few studies have examined the transformation of *B. subtilis* in biofilms. Reduced competence has been observed in static biofilms of environmental *B. subtilis* strains [198].

Most biofilm transformation studies have been conducted using streptococcal biofilms. The natural habitats of streptococci are multispecies biofilms on mucosal surfaces of humans and animals. Therefore, they are clinically relevant. The main colonizers of biofilms formed on dental surfaces belong to the *Streptococcus mitis* group, including *Streptococcus oralis*, *Streptococcus gordonii*, *Streptococcus sanguinis* and *S. mitis*, resulting in multispecies biofilms [199]. *S. pneumoniae* and *S. aureus* are common co-colonizers in the nasopharynx and oropharynx of humans [200,201,202], suggesting natural transformation in these environments. The potential of gene transfer by transformation among oral bacteria was demonstrated in static biofilms, composed of naturally transformable *S. gordonii* with either naked plasmid DNA or the plasmid-harboring spirochete *Treponema denticola* as a model exchange system [203]. The transformation rates of *S. gordonii* were determined by scraping off the biofilm and selecting transformants on antibiotic-containing agar plates.

The natural transformation efficiency of pneumococcal strains has been studied during murine nasopharyngeal co-colonization by multiple strains and with static in vitro biofilm models [172].

To study nasopharyngeal recombination upon ARG transformation, the human nasopharynx was simulated in a vertical diffusion chamber (bioreactor chamber) [204]. These chambers with pharyngeal cells were inoculated with the two pneumococcal strains to create biofilm consortia, or single-species biofilms as controls. The strains chromosomally encoded different single resistance genes. The rapid development of double antibiotic-resistant colonies was detected after 4 h of incubation within the pneumococcal biofilm consortia.

Although the natural competence of *S. mutans* planktonic cultures was demonstrated in 1981 [205], it took 20 years to demonstrate competence development in *S. mutans* biofilms [206]. Six *S. mutans* strains grown in biofilms showed 10- to 600-fold higher transformation efficiencies than their planktonic counterparts grown in liquid cultures. The natural transformation of *Streptococcus pyogenes* biofilms grown on epithelial cells has been demonstrated in vitro and in vivo during nasopharyngeal colonization in mice [207].

It was unclear for a long time whether *S. aureus* could develop natural competence. A decade ago, it was shown that a subpopulation expressing the secondary sigma factor SigH is capable of transformation under planktonic conditions [208]. However, this kind of competence requires both SigH and the SigH-dependent operons *comG* and *comE* as well as specific laboratory conditions for *sigH* expression. This suggests a need for highly specific environmental conditions for natural transformation. The development of natural transformation in static *S. aureus* biofilms was demonstrated by the transfer of the staphylococcal cassette chromosome (SCC) gene SCC*mec* [171]. Phage-dependent and conjugative transfer were excluded using phageless and heat-killed donor cells, which were unable to synthesize the conjugation apparatus.

#### Higher Transformation Rates in Early Biofilms Than in Planktonic Cultures

In summary, the higher transformation rates in biofilm communities, particularly in early biofilms, compared to planktonic cells, contribute to the spread of ARGs within biofilms. As biofilms spontaneously release planktonic cells during detachment, this could allow the dissemination of transformants to a broader range of habitats [108]. Diverse environmental and cellular cues trigger regulation of gene expression in biofilms and biofilm architecture. Multiple mechanisms and laboratory parameters (amount of exogenous DNA, choice of media, incubation time, temperature, cell density, species composition, and microbial physiological state) can induce transformation in biofilms.

Furthermore, although transformation has been studied in biofilms, in most studies, the transformation rate has not been determined *in situ*. Biofilms were disrupted instead (scraped or enzymatically removed) to enumerate transformants on selective agar plates [108,172,182,183,184,185,198,204,209,210]. The extent to which this distorted the actual transformation rate is unclear.

## 3. Transduction

Bacteriophages (phages) are viruses that infect bacteria. Transduction is the phage-mediated transfer of DNA to bacterial cells. As they are natural enemies of bacteria, they are found everywhere and are therefore widely present in natural environments. Furthermore, phages have a longer persistence compared to bacteria, and are not easily degradable under environmental conditions. They can tolerate acids, alkaline conditions or high temperatures [211] and cannot be completely eradicated by UV irradiation or chlorination [212,213]. The long-term coevolution of phages and bacteria in nature has led to complex and diverse bacteria–phage interactions and mutualistic adaptations.

Phages are non-living biological entities that exploit the replication machinery of the host to produce phage progeny. Phage-mediated transduction is initiated after phage attachment and injection of phage DNA into the bacterial cytoplasm. Lytic (virulent) phages start directly with reproduction, whereas lysogenic (temperate) phages can first integrate their genome into the host chromosome (prophage) before the lytic cycle is initiated. Bacterial DNA injected by phages must be either integrated by homologous recombination into the host chromosome or maintained as a self-replicating plasmid [214].

Phages and prophages also encode ARGs [215,216,217]. To discuss ARG transfer by transduction within biofilms, three different transduction mechanisms should be considered in more detail: generalized, specialized, and lateral transduction.

### 3.1. Generalized Transduction

The lytic cycle degrades bacterial DNA into smaller fragments. Generalized transduction is based on erroneous packaging by *pac*-type phages. Phages wrongly recognize *pac*-site homologs (pseudo*-pac* sites) encoded on heterogenous DNA, resulting in the mispackaging of random segments of bacterial DNA (plasmid or chromosomal DNA) into capsids instead of phage DNA [218,219,220]. The transfer of ARGs by generalized transducing phages has been examined in several planktonic studies, but not in biofilms [221,222,223].

### 3.2. Specialized Transduction

Specialized transduction occurs when the phage genome is incorrectly excised, together with the host DNA flanking the integrated prophage [224]. Hence, non-phage and viral DNA are erroneously encapsidated into the phage and can be transferred to a new host cell (Figure 1).

Both transduction mechanisms provide opportunities for acquiring ARGs, but with specialized transduction the variety of bacterial genes to be excised or randomly packaged is rare and limited by genes directly adjacent to the phage integration locus. Specialized transduction is probably only a minor contributor to gene (ARG) transfer compared to generalized transduction [225].

### 3.3. Lateral Transduction

One recently discovered mechanism is lateral transduction. The prophage is still integrated into its host chromosome, where bidirectional phage replication begins *in situ*. This leads to DNA encapsidation prior to excision, and replication of long fragments of adjacent bacterial DNA [226]. Phages are packed with both phage DNA and adjacent bacterial genes. Up to 100 kb of bacterial DNA downstream of the prophage can be encapsidated [227]. The immense additional excision of genes may offer potential transfer routes for resistance genes. In contrast to specialized transduction, lateral transduction is not based on incorrect excision but rather on another natural mode of phage transduction. Lateral transduction was first shown for lysogenic *S. aureus* phages [226], but also for *P. aeruginosa* [228], and *S. enterica* [229]. All examined strains were characterized as biofilm formers, so it can be assumed that this transduction mechanism takes place in biofilms as well. To the best of our knowledge, no biofilm studies on lateral transduction have been published so far.

### 3.4. Lack of Transduction Experiments in Biofilms

Transduction experiments have commonly been performed using planktonic cultures with characterized environmental, clinical, or laboratory strains [230,231,232,233]. To the best of our knowledge, only one study has demonstrated phage-mediated gene transfer through lysogenic bacteriophages in biofilms of potentially pathogenic *E. coli* [234]. In this study, recipient biofilms were formed on glass slides under static conditions. The donor strain carrying a Shiga-toxin (Stx)-encoding bacteriophage was added to the predeveloped recipient biofilm and both were incubated together. The biofilm was scraped off, disrupted, and plated onto selective agar to enumerate the transductants. The number of transductants increased over time, and donor cells were no longer detectable after six days of cultivation. The authors hypothesized that donor cells were lysed after entering the recipient biofilm. This study demonstrated that gene transfer via transduction can occur in biofilms.

### 3.5. Potential of Phages in ARG Dissemination

#### 3.5.1. Occurrence of Phages in Aquatic Environments

Because phages are non-motile, bacterial host infections are mainly based on the passive diffusion of phages within biofilms [235,236]. The transduction of DNA, and thus ARGs, is further limited by their non-motility. The total number of phage particles on Earth is assumed to be close to 10^31^ [237,238]. Phage density is estimated to be approximately 10^6^/mL in coastal marine waters [239]. Infecting bacterial hosts in ecological niches, such as aquatic environments, is feasible because of high phage concentrations. Furthermore, it has been recently shown that coliphages adsorbed onto the flagellated nonhost *Bacillus cereus* act as carriers for facilitated phage migration into aquatic *E. coli* biofilms [240]. Early *in situ* field studies have shown generalized transduction between *P. aeruginosa* strains in aquatic environments [241,242,243,244]. Accordingly, studies on the impact of phages on the spread of resistance genes are still of interest.

#### 3.5.2. Phage Host Range and Its Contribution to the Spread of ARGs

Transduction events are limited by the species or even strain specificity of bacteriophages, which in turn makes the mechanism less important for the spread of ARGs than transformation and conjugation [224,245]. Nonetheless, phages represent vehicles for the dissemination of antibiotic resistance [246]. Furthermore, a few phages have been shown to infect a broad range of species [247]. Some phages even cross genera barriers [248].

Studies on phages as ARG reservoirs are controversial, and phage-associated ARGs are assumed to be rare [249,250,251]. On the other hand, phages recovered from various environments carry functional and novel ARGs. Therefore, they can represent reservoirs of ARGs which are disseminated via phage–host interactions [216,252,253].

Phage-mediated transduction has been studied in detail in a wide range of genera, including *Staphylococcus*, *Pseudomonas*, *Acinetobacter, Clostridium* and *Salmonella* [230,254,255,256]. Transduction is assumed to be the driving force of HGT in *S. aureus* [257,258,259,260]. To the best of our knowledge, no studies on transduction events of clinically relevant Staphylococci, such as CoNS or *S. aureus,* in biofilms have been published to date.

### 3.6. Challenges of Studying Transduction in Biofilms

Biofilms are less susceptible to phage penetration than planktonic cultures because they have developed a broad spectrum of defense mechanisms against phage predation. Suitable characteristics and defense mechanisms to protect against phages include diffusion inhibition, biofilm thickness, metabolic state of bacteria (slow growth, low/no metabolic activity), biofilm maturity level, phage adsorption by the biofilm matrix, and phage-resistant/non-susceptible bacteria [261].

The effects of phage-mediated transfer and phage-bacteria interactions in many ecosystems remain poorly characterized. It is assumed that transduction efficiencies in biofilms differ from those in planktonic cultures. Bacterial cultures in a well-mixed broth allow a random distribution. In biofilms, phages first adsorb to a single bacterium, and subsequently infect adjacent cells. However, it is less likely that they infect more distant bacteria [214]. Suitable biofilm models should be established to monitor transduction not only in planktonic cultures.

Many phage–host interactions have not been studied so far due to the non-cultivability of host bacteria and difficulties in phage isolation and detection [262,263]. Transduction should be detected not only by classical methods such as conventional plating using selectable genetic markers. State-of-the-art imaging techniques, such as epifluorescence microscopy or atomic force microscopy have been applied to detect and characterize uncultivated phages in environmental samples, as recently reviewed by Turzynski et al. [264].

A labelling technique to quantify transduction at single-cell level is the so-called cycled primed *in situ* amplification-fluorescent *in situ* hybridization (CPRINS-FISH) [265,266]. This method combines the features of fluorescence *in situ* hybridization (FISH) and *in situ* PCR.

Although the range of methodologies has expanded over recent years, biofilm studies with phages have focused more on the potential of lytic bacteriophages as antibiofilm agents or as an alternative treatment of antibiotic-resistant bacterial infections [267].

## 4. Conjugation

Conjugation is a major mechanism that facilitates bacterial adaptation by allowing the genetic exchange of ARGs, resistance to heavy metals, virulence, and other advantageous properties. It occurs at higher rates in biofilms than in planktonic cultures [139,268,269]. In contrast to transduction and transformation, conjugation is based on direct cell-to-cell contact and mating pair formation between a donor and recipient cell established by adhesins in Gram-positive and conjugative pili in Gram-negative bacteria, as these two cell types fundamentally differ in their cell wall structure. A multiprotein complex spans the cell envelope and mediates unidirectional DNA transfer from a donor to a recipient cell. The conjugation process is mediated by MGEs such as conjugative plasmids, integrative and conjugative elements (ICEs), or transposons. Gram-negative conjugation has been studied in more detail [270,271,272,273,274]. Several techniques are available to track the transfer of MGEs in microbial biofilm communities and planktonic cultures, which are addressed in the following paragraphs. Demonstrating conjugative transfer with planktonic cultures prior to biofilm studies can serve as a reference for transconjugant detection and assessment of MGE transferability. Mating assays are conventionally performed to verify the conjugative transfer of MGEs. They are mostly performed as liquid, solid, or filter mating assays. Conjugative transfer is experimentally measured as the transfer rate (transconjugants per donor or recipient cell) [128,275,276,277,278,279,280]. Solid and filter matings seem to be the most suitable methods, resulting in higher transfer rates in comparison to liquid mating [280,281,282,283]. This could be explained by the similarity to natural conditions using solid surfaces, increased heterogeneity, and spatial structures for surface-associated microbes and, therefore, microbial communities. However, in vitro mating assays are insufficient to fully understand in vivo circumstances within natural biofilms and possess certain limitations in determining the natural transfer rate of MGEs.

Most conjugation studies have been performed with plasmids, but in recent years, other MGEs, such as ICEs and integrons, have also come into focus. The following chapter points out the current knowledge about conjugation in biofilms referring to different conjugative elements.

Integrons and genomic islands (chromosomal, resistance, or pathogenicity islands) also contribute to the distribution of ARGs via HGT [284,285,286,287]. These elements are not covered in detail here due to scarce data on biofilm studies. This chapter focuses on conjugative plasmids and ICEs instead.

### 4.1. Conjugative Plasmids

Plasmids autonomously replicate as extra-chromosomal elements that do not carry genes that are essential for bacterial survival. Conjugative plasmids harbor genes for a T4SS. Mobilizable (non-conjugative) plasmids do not encode the entire conjugation machinery. Instead, they require conjugative elements in the same cell to be transferred to a new host [288]. They carry at least an origin of transfer (*oriT*) sequence and a gene encoding the relaxase protein, which nicks the plasmid DNA at the *oriT* site, thus initiating the transport of ssDNA to the T4SS provided by a conjugative plasmid [289]. Plasmids carry not only transfer genes but are important vehicles for single or multiple ARGs even against “last resort antibiotics” [290,291,292,293]. Plasmids can naturally transfer between different genera or phyla [272,294,295]. Figure 1 shows conjugative plasmid transfer for both Gram-negative and Gram-positive T4SS.

Thus, the host range of conjugative plasmids greatly contributes to the spread of ARGs. As many broad-host-range conjugative plasmids are of clinical origin [272,296,297,298], their release and distribution in the environment should be monitored.

Even though studying HGT in microbial communities can be less invasive, biofilms are often disrupted for further analyses such as staining and subsequent visualization or determination of conjugative transfer rates by plating [299]. The transfer of the broad-host-range plasmid RP4 was demonstrated in a mono-species biofilm of *Pseudomonas* strains by cultivating them in a rotating annular reactor (bioreactor) [280]. The conjugative transfer rate was determined with plating of disrupted biofilm samples. Another study used four different growth systems (three types of flow cells and a biofilm reactor) for cultivation and plate counting to quantify IncP-1 pB10 plasmid transfer in *E. coli* biofilms [128]. However, transfer rates in detached and disrupted biofilms are not comparable to those in intact biofilms. The location of transfer events can also not be determined. *In situ* biofilm formation is a promising variant to overcome many obstacles in the experimental design of environmental biofilm studies. However, it is often not used due to the experimental complexity and the use of invasive detection methods. Even when conjugation experiments occur under biofilm conditions, many researchers still use plates to enumerate transconjugants [128,138,139,140,141,300,301,302].

#### Methodology to Assess Conjugative Plasmid Transfer Rates *in situ*

Beaudoin et al. demonstrated that the conjugative transfer rate can be determined *in situ* [137]. After biofilm cultivation in a bioreactor, cryo-embedded and cryo-sectioned biofilm slides were stained to microscopically visualize donors and transconjugants expressing β-galactosidase (from a *lacZ*-containing mobilizable plasmid). Transconjugants were further distinguished by their relative cell sizes, with *P. putida* as the donor and *Bacillus azotoformas* as the recipient. The number of transconjugants was given per unit biofilm surface [cells/m^2^/h].

CLSM can be used as a non-invasive technology to directly monitor HGT in biofilms [128,135,155,187,189,303,304,305]. The milestone in monitoring conjugation in complex microbial communities was the use of a single GFP reporter as a fluorescent plasmid marker and chromosomally encoded luciferase genes (bioluminescence) for recipient labelling [306]. In this study, conjugative dissemination of the TOL plasmid was not investigated in biofilms, but on nutrient agar plates. In other studies, GFP expression was repressed in *P. putida* donor cells carrying a reporter plasmid whose fluorescence was repressed by a chromosomal repressor (*lacI*^q^). Therefore, only transconjugant cells were fluorescent [127,305,307]. *In situ* determination of conjugative transfer rates in defined biofilms with GFP labelling was performed by Hausner and Wuertz [303]. Transfer rates were given as ratios of transconjugants to recipient cell per hour of contact time [number of transconjugants/recipient cell/h]. The conjugation rates were 1000-fold higher than those determined by solid surface mating.

An optimized technique for studying conjugation *in situ* is a dual-label monitoring tool that uses fluorescent reporter genes without any additional stains [304,308]. This technique was utilized to study environmental communities, such as wastewater and soil. The expression of two fluorescent proteins enabled tracking of the donor (chromosomal *dsRed* gene) and the conjugative TOL plasmid (*gfp* gene) for *in situ* monitoring of plasmid transfer in biofilm reactors [155,304,309]. Klümper and coworkers used a donor chromosomally tagged with a constitutive red fluorescence reporter gene and a plasmid of interest encoding *gfp.* The GFP expression was controlled by a *lacI*^q^-repressible promoter and LacI production in the donor [308]. Due to the absence of a *lacI*^q^-repressible promoter in the recipients, *gfp* expression was restored. In this study, *E. coli*, *P. putida* and *Kluyvera* sp. strains were used as donors and a soil microbial community as recipients. Thus, the differentiation of donor, recipient, and transconjugant cells is highly specific.

There exist several challenges when applying the presented methods to biofilms: Bacteria within natural biofilms underlie local gradients of abiotic factors (nutrients, oxygen, pH, and waste products) [310]. GFP requires oxygen for maturation and activation [311,312]. With increasing biofilm depth or high density in biofilms, interior oxygen availability decreases and thus impairs GFP expression [269]. Anaerobic liquid cultures of *E. coli* were subjected to varying acidic pH values and simultaneously used for the anaerobic expression of fluorescent proteins mCherry and GFPmut3 [123]. The fluorescence of both proteins was completely recovered after exposure to oxygen; this is the so-called aerobic fluorescence recovery. At pH <5 only the fluorescence of GFPmut3 was fully recovered. Low pH values might lead to protein degradation and thus irreversible quenching of fluorescence.

A promising platform for studying HGT in biofilms is microfluidics. The technique allows on-chip mating assays for plasmid-mediated conjugation in biofilms at a microscale level. This method enables long-term cultivation, time-lapse and real-time tracking of transfer processes, transconjugant formation even at the single-cell level, and real-time imaging of *in situ* biofilm formation [142,313]. In the study of Li and coworkers, the *E. coli* donor strain carried the GFP-encoding IncP-1 plasmid pKJK5 with dual-label fluorescence [313]. The donor was cultivated together with either activated sludge community recipients or single-strain recipients on the chip. Microfluidics can mimic natural biofilms more promisingly and lead to a better understanding of the environmental influences on HGT compared to end-point measurements from solid surface matings [142,294,314]. Conjugation of the fluorescently labeled broad-host-range IncP-1 plasmid RP4 has been monitored in real time in single-species and mixed environmental biofilms using microfluidics [142,294,314].

Fluorescent labeling methods can be combined to specifically detect donors, recipients and transconjugants. In contrast to in vitro HGT determination, time-lapse imaging using microfluidics can further distinguish between vertical transmission and HGT. Randomly appearing GFP spots can be described as primary HGT events, the fluorescence extension from these spots is due to cell division [313].

However, there are also some drawbacks of the system: potential instability of GFP-encoding plasmids in transconjugants or repression of *gfp* expression by the natural *lac* operon in *E. coli* recipients [307] or autofluorescence of uncharacterized recipients [315] cannot be excluded.

### 4.2. Integrative and Conjugative Elements (ICEs)

ICEs, formerly termed conjugative transposons, are MGEs that encode genes for their own excision, conjugation, and integration [316]. ICEs are mostly hidden in bacterial chromosomes. Thus, they are passively inherited vertically but can move horizontally by conjugation via self-encoded T4SSs between donor and recipient strains (Figure 1). Other MGEs, such as integrative and mobilizable elements (IMEs), encode genes for their excision and integration. Due to the lack of genes for conjugative transfer, IMEs must hijack the conjugative apparatus of co-residing self-transmissible elements, such as ICEs. IMEs carry ARGs and may play a major role in the dissemination of ARGs [317,318]. Identifying and annotating new putative ICEs and IMEs in bacterial chromosomes in silico remains challenging [319,320]. Nonetheless, ICEs and IMEs have been suggested to be more phylogenetically widespread than conjugative plasmids [316,318,321].

As IMEs are rarely described, no HGT studies in biofilms exist to date.

ICEs can exist in circular extrachromosomal forms upon excision and transfer to a new host (Botelho, 2020). However, they are not stable as an extrachromosomal intermediate. They replicate only when integrated into the host chromosome via site-specific recombination between defined attachment (*att*) sites [322]. The ICE life cycle, excision, and integration have been extensively reviewed in recent years [86,323,324]. ICEs often carry cargo genes including metabolic pathways, determinants of pathogenesis and symbiosis, and ARGs [86,324,325].

There is little knowledge on ICEs compared to plasmids. A review from 2020 compared PubMed searches with combined keywords such as “antibiotic resistances” AND “integrative conjugative element” or AND “conjugative transposon” with that of “plasmid” AND “antibiotic resistance” [324]. Only 2–7% hits were found compared to a search combining the keywords “antibiotic resistance” AND “plasmid”. There are just a handful of studies that investigated ICEs in biofilms. Nevertheless, multiple studies have demonstrated ICEs as the main reservoir for several resistance genes in biofilm-producing species [326,327,328,329,330].

Evolutionary dynamics and the mechanism of ICE conjugation remain largely unexplored [324]. Although many potential ICEs have been found in recent decades, only a few well-characterized ICE models are available to study propagation and transfer. These include *B. subtilis* ICE*Bs1* [331], *V. cholerae* ICE*SXT*-R391 [332], *E. faecalis* Tn*916* [333]*, Bacteroides thetaiotaomicron* CTn*DOT* [334], *Pseudomonas knackmussii* ICE*clc*, *Streptococcus thermophilus* ICE*St1/3* [335,336], and *Mesorhizobium loti* ICE*MlSym*^R7A^ [337]. The genetic organization and main regulatory networks controlling ICE transfer were reviewed in detail in 2017 [323]. To improve their trackability, resistance markers have been inserted into ICE*St1* and ICE*St3* [338] as well as into ICE*Bs1* [299,339].

#### Methodology to Assess ICE Transfer Rates in Biofilms

Conjugation studies in biofilms have only been performed with ICE*Bs1*. They were compared with solid surface matings. The donor–recipient mixture was incubated on biofilm-stimulating growth medium to induce biofilm formation. The same experimental setting has been used in several ICE*Bs1* studies [299,339,340]. In comparison to non-inducing media, a 100- to 10,000-fold increase in ICE transfer was demonstrated [339]. In addition, ICE*Bs1* transfer frequency was increased by extracellular matrix production by the recipient cells.

ICE dissemination in *B. subtilis* biofilms was analyzed by cryomicrotome and fluorescence microscopy [340]. High ICE*Bs1* transfer rates were almost exclusively driven by transconjugants as secondary transfer events. HGT was mainly mediated by transconjugants in close vicinity to initial donor cells, in so-called conjugative clusters, located close to the air–biofilm interface.

It is often not possible to determine exactly the impact of vertical transfer on the dissemination of MGEs within the biofilm. To this end, Bourassa et al. [340] deleted an essential gene of the ICE*Bs1* T4SS machinery. The knockout was only complemented *in trans* in the donor. Thus, transconjugants cannot transfer ICEs via conjugation.

As in the conjugation experiments of Klümper et al. [308], fluorescence of transconjugants was controlled by the Lac repressor. Donor–recipient mixtures were cultivated on agarose slices to facilitate mating. ICE*Bs1* transfer in *B. subtilis* cells was visualized in real time by fluorescence microscopy [341]. Red fluorescence was expressed in donor cells. GFP expression was repressed by fusing the *gfp* gene to the *E. coli* Lac repressor *lacI*^q^ (LacI-GFP) inserted into ICE*Bs1*. Thus, *B. subtilis* cells which had acquired ICE*Bs1* showed green fluorescence*. B. subtilis* cells grow naturally in chains. The authors demonstrated that rapid ICE*Bs1* spread among cells within the bacterial cell chain occurred via ICE conjugation.

To follow ICE transfer in real time, this fluorescent method could be applied to biofilm models, such as flow chambers or microfluidics. In comparison to plasmids, more extensive studies are needed to better validate the role of ICEs in biofilms.

## 5. Conclusions and Perspectives

This review aimed to provide deeper insights into the ability of pathogens to acquire ARGs and adapt to changing environments. Environmental and clinical biofilms are reservoirs of ARG-encoding bacteria. They are a serious threat to public health, as they can intervene with the treatment of infectious diseases. HGT mechanisms, such as transformation, transduction, and conjugation, shape the genetic diversity of bacteria and contribute to their rapid adaptive evolution. Mimicking biofilm conditions facilitates studying the spread of ARGs across different ecological systems. HGT experiments in biofilms are still underrepresented as biofilms are highly complex structures that are difficult to mimic under laboratory conditions. Thus, multiple approaches are required to study horizontal gene spread in biofilms. Most biofilm studies still use invasive methods to determine HGT rates. Coupling classical methods with *in situ* techniques such as CLSM or microfluidics would immensely enhance the knowledge of gene spread in complex environments. The use of modern and more accurate methods such as Hi-C or CRISPR–Cas spacer acquisition can help overcome the insufficient linkage of MGEs to their hosts and the lack of detection of transient HGT events. In addition, coupling biofilm studies with the application of modern HGT detection and quantification methods would help identify risk scenarios to develop intervention strategies against ARG spread.

The pool of MGEs encoding ARGs is immense. However, the transfer features of OMVs, MVs, ICEs, IMEs, integrons and other mobile gene cassettes are not as well characterized as those of conjugative plasmids. The dissemination potential of these mobile elements in heterogeneous environments such as biofilms should not be underestimated and should be studied to a greater degree in the future.

In summary, biofilms should be an important target for regular monitoring and control of the occurrence and dissemination of ARGs in clinical settings, industry, and the environment.

## Figures and Tables

**Figure 1 antibiotics-12-00328-f001:**
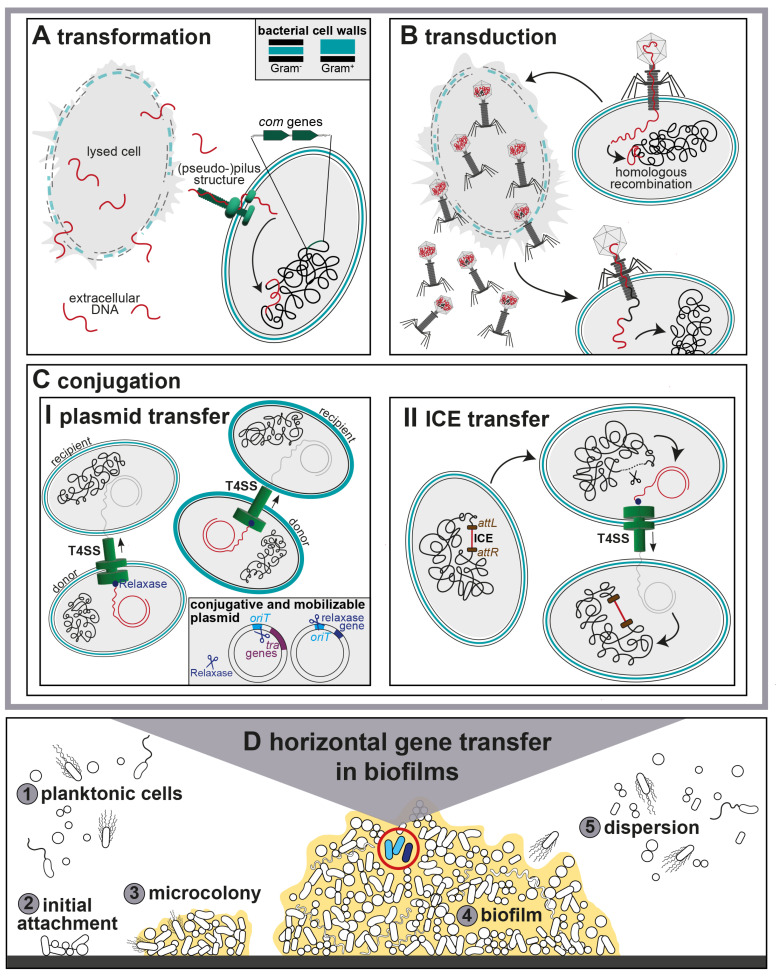
Horizontal gene transfer in polymicrobial biofilms. The upper panels **A**–**C** show the three main HGT mechanisms (**A**) Transformation, (**B**) Transduction and (**C**) Conjugation. Transferred, transduced or conjugative DNA is shown in red, host DNA in black. (**A**) Transformation of competent Gram-negative bacteria. Uptake of extracellular DNA with pilus-like structures followed by incorporation into the host genome by homologous recombination. Chromosomally encoded competence (*com*) genes are marked in green. (**B**) Specialized transduction starts with the injection of phage DNA and integration into the host chromosome. Phage progeny is released during the lytic cycle. Phages can contain both viral and chromosomal DNA by erroneous encapsidation. This can be transferred to a new host cell. (**C**) Conjugation is illustrated in two ways: (I) Conjugative plasmid transfer for both Gram-negative and Gram-positive Type IV Secretion systems (T4SS). The relaxase (blue) bound to the single-stranded (ss) DNA after cleaving the plasmid at the origin of transfer (*oriT*) prior to transfer. Conjugative plasmids contain *oriT* and transfer (*tra*) genes, mobilizable plasmids only *oriT* and relaxase genes. (II) ICE transfer is introduced by excision of the ICE from the host chromosome at attachment (*att*) sites (brown). (**D**) Stages of polymicrobial biofilm formation from (1,2) initial attachment of planktonic cells, (3) microcolony formation to (5) dispersion of cells from a (4) mature biofilm. The light blue cells (4) show a conjugation event, the dark blue cell transformation or transduction.

## Data Availability

Not applicable.

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
