# Peer review of "Horizontal Gene Transfer of Antibiotic Resistance Genes in Biofilms"

_antibiotics, 2023, doi:10.3390/antibiotics12020328_

Round 1

Reviewer 1 Report

The manuscript "Horizontal gene transfer of antibiotic resistance genes in biofilms" has an interesting outcome. However, several reviews have very recently addressed the same topic and it is hard to see the novelty of this particular review and its benefit to Antibiotics readership. However, for publication in Antibiotics, the manuscript needs to be improved.

  1. L39 and 40: Use the complete form of all abbreviated words for the first time throughout the manuscript.
  2. There are several environmental factors involved in biofilm formation. But this information is missing in the review. Authors should add this information with proper references.
  3. L55-57: Why do you think only these three Gram-negative and two Gram-positive bacteria are involved in biofilm-associated infections? What about other pathogens? Authors should explain more elaborately about biofilm-associated diseases in humans.
  4. Authors should use the most recent references (last 5 years) throughout the manuscript.

Author Response

  1. The manuscript "Horizontal gene transfer of antibiotic resistance genes in biofilms" has an interesting outcome. However, several reviews have very recently addressed the same topic and it is hard to see the novelty of this particular review and its benefit to Antibiotics readership. However, for publication in Antibiotics, the manuscript needs to be improved.

Response: Thank you for the comment.

Compared to previous reviews about HGT in biofilms:

  • We clearly distinguished between research on biofilms from gram-positive and gram-negative bacteria.
  • We accentuated studies on conjugative transfer in gram-positive bacteria.
  • In addition, a focus was laid on conjugative plasmids AND ICEs.
  • By dividing our chapters into HGT studies in biofilms from gram-positive and gram-negative bacteria, we showed the lack of knowledge about gram-positive HGT.
  • We highlighted the mechanistic difference between conjugative transfer in gram-negative and gram-positive bacteria by presenting a detailed figure showing the differences of the respective type IV secretion systems (Fig. 1).
  • We fully updated the literature about membrane vesicle-mediated transfer.
  • In addition, we presented major gaps of knowledge in the field, e.g., the lack of transduction experiments in biofilms.
  • We critically evaluated the methodology to measure HGT mediated by transformation, transduction, and conjugation in biofilms.
  1. L39 and 40: Use the complete form of all abbreviated words for the first time throughout the manuscript.

Response: Thank you for this comment.

The complete form of horizontal gene transfer and antibiotic resistance genes was inserted in the main manuscript when it was used for the first time.

  1. There are several environmental factors involved in biofilm formation. But this information is missing in the review. Authors should add this information with proper references.

Response: We added an additional chapter 1.5 “Environmental factors influencing biofilm formation” (lines 234-258 in the manuscript with track changes) to underline how environmental factors are involved in biofilm formation. Recent reviews and publications from the last years were included.

  1. L55-57: Why do you think only these three Gram-negative and two Gram-positive bacteria are involved in biofilm-associated infections? What about other pathogens? Authors should explain more elaborately about biofilm-associated diseases in humans.

Response: Thank you very much for pointing this out. We added further relevant bacteria involved in biofilm-associated infections (lines 104-114 in the manuscript with track changes). We also added information about biofilm-associated diseases in humans (lines 115-122).

 5. Authors should use the most recent references (last 5 years) throughout the manuscript.

Response: We updated the literature throughout the manuscript and added new paragraphs and chapters (we added approximately 60 new references). In some areas such as transformation or transduction in biofilms only few studies have been published in recent years. The respective gap of knowledge is mentioned within the text. To the best of our knowledge, all important new evidence and research in the area has been incorporated into the manuscript.

Reviewer 2 Report

I highly appreciate the topic of this review and the review itself. The topic of Horizontal gene transfer of antibiotic resistance genes in biofilms is highly actual, thus to review the yet known knowledge is highly important. The review is well structurised and readable, giving very good insight into the problematic. I do not have any remarks and I recommend to publish it in present form.

Author Response

We kindly thank you for the positive feedback.

Reviewer 3 Report

The authors have reported a comprehensive collection of biofilm evidence in this manuscript. The topics addressed concern: the formation of biofilms on biotic and abiotic surfaces, the clinical impact of biofilms in the treatment of infections with antibiotics, and the transfer of resistance genes between bacterial cells present in the matrix. The authors described each paragraph well, and the manuscript could be revised after a few revisions noted below.

1. Introduction paragraph: Authors should well incorporate the current and accredited definition of biofilm

2. The authors could, to provide further information about the manuscript, insert the paragraph on methodologies (See review reference 56)

3. Section: 1.1 Biofilm development: the authors could well describe the four stages of biofilm formation by means of a bullet list (1. Planktonic cells, 2. Initial attachment, 3. Microcolony, 4. Mature biofilm 5. Dispersion)

4. biofilm on medical devices: The authors should better describe this aspect, as the contamination of medical devices is a serious problem for the management of MDR-microorganism pathologies. By implementing this section, the review would acquire an important impact from a clinical point of view. An article by Folliero et. describes the correlation between biofilm formation on medical devices and the acquisition of multidrug resistance.

5. Section 1.3 Horizontal gene transfer in biofilms: replacing membrane vesicles directly with outer membrane vesicles (OMVs)

6. Section 1.3 Horizontal gene transfer in biofilms: in addition to reference 56, another study by dell'Annunziata et. to the. describes the role of OMVs on HGT mediated by K. pneumoniae, an important opportunistic pathogen recognized to be a strong biofilm producer.

7. Regarding OMVs, these vesicles are known to increase biofilm production in many bacterial strains, such as P.aeruginosa. In my opinion, this is an important aspect that has not been studied so far and could give added value to the manuscript.

8. Transformation-Transduction-Conjugation: The authors should, before each paragraph, insert a summary table of the experimental evidence reported in the text.

Author Response

The authors have reported a comprehensive collection of biofilm evidence in this manuscript. The topics addressed concern: the formation of biofilms on biotic and abiotic surfaces, the clinical impact of biofilms in the treatment of infections with antibiotics, and the transfer of resistance genes between bacterial cells present in the matrix. The authors described each paragraph well, and the manuscript could be revised after a few revisions noted below.

1. Introduction paragraph: Authors should well incorporate the current and accredited definition of biofilm

Response: Thank you for the comment. We added a recent definition of biofilms in the introduction (lines 27-32 in the manuscript with track changes).

2. The authors could, to provide further information about the manuscript, insert the paragraph on methodologies (See review reference 56)

Response: Thank you for this remark. We added a new chapter 1.1 on methodologies which describes the main focus of the review as well as the methods applied to compile it.

3. Section: 1.1 Biofilm development: the authors could well describe the four stages of biofilm formation by means of a bullet list (1. Planktonic cells, 2. Initial attachment, 3. Microcolony, 4. Mature biofilm 5. Dispersion)

Response: Thank you for pointing this out.

We added the missing information in the main text in chapter 1.2 (lines 72-78 in the manuscript with track changes). Furthermore, we moved the paragraph to the beginning of the chapter.

4. biofilm on medical devices: The authors should better describe this aspect, as the contamination of medical devices is a serious problem for the management of MDR-microorganism pathologies. By implementing this section, the review would acquire an important impact from a clinical point of view. An article by Folliero et. describes the correlation between biofilm formation on medical devices and the acquisition of multidrug resistance.

Response: Thank you for the valuable comment. We added a new paragraph about medical biofilms in chapter 1.2. We pointed out (1) the occurrence of biofilms in the medical environment, (2) gave examples for biofilm-related infections, (3) gave examples of multidrug-resistant biofilm formers in the clinical environment and (4) connected these important aspects with the topic of the review “horizontal gene transfer in biofilms” (lines 104-122 in the manuscript with track changes).

5. Section 1.3 Horizontal gene transfer in biofilms: replacing membrane vesicles directly with outer membrane vesicles (OMVs)

Response: Thank you for the comment. We also included a definition of membrane vesicles in gram-positive and gram-negative bacteria (lines 191-195).

6. Section 1.3 Horizontal gene transfer in biofilms: in addition to reference 56, another study by dell'Annunziata et al to the describes the role of OMVs on HGT mediated by K.  pneumoniae, an important opportunistic pathogen recognized to be a strong biofilm producer.

Response: Thank you for the valuable comment. We added some new references and a new paragraph in chapter 1.4 (lines 195-206) which describes MVs and OMVs in more detail. Furthermore, we included OMVs and MVs in chapter 5 “conclusion and perspectives” (line 881 in the manuscript with track changes).

7. Regarding OMVs, these vesicles are known to increase biofilm production in many bacterial strains, such as P. aeruginosa. In my opinion, this is an important aspect that has not been studied so far and could give added value to the manuscript.

Response: Thank you for pointing this out. We added these important aspects in the new paragraph in chapter 1.4. (lines 211-220 in the manuscript with track changes).

8. Transformation-Transduction-Conjugation: The authors should, before each paragraph, insert a summary table of the experimental evidence reported in the text.

Response: Thank you for this remark. Instead of summary tables, we decided to include additional informative subheadings (2.1.1., 2.2.1., 4.1.1., 4.2.1.) into the chapters to improve the structure of the text and facilitate reading.